# COVID-19-Induced Reduction in Primary Melanoma Diagnoses: Experience from a Dermatopathology Referral Center

**DOI:** 10.3390/jcm10184059

**Published:** 2021-09-08

**Authors:** Magdalena Hoellwerth, Andreas Kaiser, Michael Emberger, Matthias Brandlmaier, Martin Laimer, Alexander Egger, Johann W. Bauer, Peter Koelblinger

**Affiliations:** 1Department of Dermatology and Allergology, Paracelsus Medical University, 5020 Salzburg, Austria; m.hoellwerth@salk.at (M.H.); m.brandlmaier@salk.at (M.B.); m.laimer@salk.at (M.L.); a.egger@salk.at (A.E.); joh.bauer@salk.at (J.W.B.); 2Department of Clinical Psychology, Paracelsus Medical University, 5020 Salzburg, Austria; a.kaiser@salk.at; 3Patholab Salzburg, Emberger/Woelfl/Bogner OG, Labor für Pathologie, 5020 Salzburg, Austria; emberger@pathologie-salzburg.at

**Keywords:** melanoma, COVID-19, pandemic

## Abstract

The collateral damage caused by COVID-19 pandemic-associated public health and governmental measures on patient care has been increasingly assessed in various oncological and non-oncological clinical settings. In order to investigate potential adverse effects in the field of melanoma the present study analyzed the characteristics of primary melanoma diagnoses at an Austrian dermato-pathological referral center before, during, and after the first coronavirus-related lockdown in March 2020. As suspected, we found significant temporary reductions in the number of newly diagnosed melanomas in 2020 compared to previous years, in particular, during the first lockdown period.

## 1. Introduction

On 11 March 2020, the World Health Organization declared the coronavirus disease 2019 (COVID-19) a global pandemic. Subsequently, a broad range of different measures were implemented worldwide to contain further virus spread. Several governments and national authorities imposed so called shut- and lockdowns, during which the population was advised to avoid non-urgent doctor visits.

Apart from serious economic and social consequences, the adverse impact of these regulations on health care standards has become apparent in different sectors of national health care systems and across various medical disciplines, including those that are not primarily involved in the management of COVID-19 patients and thus are collaterally affected.

In the beginning of the pandemic, for instance, a major reduction in the number of cardiovascular hospitalizations and associated treatment delays of myocardial infarction were reported [1,2]. Likewise, a substantial decline in screening examinations (including those for skin cancer) was observed. Consistently, preliminary data revealed a remarkable fall in the number of diagnoses of different cancer entities in April 2020 compared to the same period of the previous year [3].

Malignant melanoma is the deadliest form of skin cancer. The vertical (Breslow) tumor thickness represents the most important prognostic criterion when evaluating primary tumors [4]. Increased Breslow thickness and tumor ulceration indicate an aggressive course and correlate with a poor prognosis owing to a higher risk of metastasis. Therefore, the early detection and excision of primary melanomas are essential [5].

Assessing the pandemic’s impact on melanoma diagnosis, Ricci et al. observed an increase in the mean Breslow thickness of primary tumors diagnosed in an Italian cohort subsequent to the first COVID-19 lockdown [6].

Against this background, our study was intended to elucidate potential changes in the number and characteristics of primary melanomas diagnosed at our dermatopathology laboratory before, during, and after the first COVID-19-related lockdown period.

## 2. Materials and Methods

After approval by the local ethics committee, data were retrospectively collected at our dermatopathology referral center, which analyses the majority of skin tissue samples obtained by dermatologists (and general practitioners) within the state of Salzburg and neighboring areas. This institution diagnoses about 14% of the approximately 3300 primary melanomas nationally recorded per year [7] and provided full service and unrestricted diagnostic capacities during the lockdown.

Collected data comprised patient age, sex, melanoma Breslow thickness, and ulceration status. Cohorts were stratified through division into seven different time periods, in order to compare the gradual effects of the lockdown in a targeted manner, each lasting 28 days: pre-lockdown period I (11 January 2020–8 February 2020), pre-lockdown period II (9 February 2020–8 March 2020), lockdown period I (16 March 2020–13 April 2020), post-lockdown period I (14 April 2020–12 May 2020), post-lockdown period II (13 May 2020–10 June 2020), post-lockdown period III (11 June 2020–9 July 2020) and lockdown II (10 November 2020–8 December 2020). To enable comparison, the corresponding time periods of the two years prior to the pandemic (2018 and 2019) were equally assessed.

We determined the median Breslow index and the percentage of ulcerated and non-ulcerated primary tumors for each time interval. The numbers of melanoma cases diagnosed within each 28 day period were analyzed and compared by the use of univariate statistical methods. The ulceration status of the melanomas was compared between the analyzed time periods using the Chi-quadrat test. All statistical tests were two-sided with the significance level at 0.05. Data processing was performed with Excel 2016 for Microsoft Windows^®^ (Microsoft Corporation, Redmond, WA, USA). For all statistical calculations, IBM SPSS Statistics 24^®^ (IBM Corp., Armonk, NY, USA) was used.

In order to evaluate a potential effect of the lockdown measures on our inpatient service, we additionally collected the number of sentinel lymph node biopsies for melanoma at the University Hospital Salzburg in the years 2018, 2019, and 2020. In our institution, this procedure is regularly carried out in an inpatient setting and generally recommended for patients with primary melanomas of more than 1 mm in Breslow thickness [8].

## 3. Results

In total, 428, 505, and 432 primary melanomas were diagnosed in 2018, 2019, and 2020, respectively (Table 1).

Comparing distinct time periods, as defined above, we found significant differences in the number of melanoma cases between 2018 and 2020 (*p* < 0.001) as well as between 2019 and 2020 (*p* = 0.022); i.e., differences were detected between 2018 and 2020 for the first pre-lockdown period I (11 January–8 February), lockdown period I (16 March–13 April), and post-lockdown period I (14 April–12 May). Between 2019 and 2020, a significant difference was found only regarding lockdown period I (16 March–13 April). As depicted in Figure 1 and Figure 2, during lockdown period I (16 March–13 April), 32 and 43 melanomas were diagnosed in 2018 and 2019, respectively, compared to only 18 melanomas in 2020.

In the period subsequent to the lockdown in 2020 (post-lockdown period I), the number of melanoma diagnoses continued to decrease compared to previous years (2018:44; 2019:21; 2020:19). Later on in 2020, the number of diagnoses almost doubled in post-lockdown period II in relation to lockdown period I (18 in lockdown period I in 2020 vs. 37 in post-lockdown period II in 2020). In Figure 3, the described trends are additionally depicted on a per week basis, confirming a marked decrease in weekly melanoma diagnoses shortly after the first lockdown measures were implemented in Austria in March 2020.

Although the mean Breslow thickness increased slightly after the first COVID-19 lockdown in 2020 (0.76 mm in lockdown period I vs. 0.83 mm in post-lockdown period I), significant alterations in Breslow thickness were neither observed between the different periods of 2020 nor in comparison to previous years.

Regarding the primary tumor ulceration status, the proportion of ulcerated melanomas was 8.3 percent in the year 2020 (*n* = 36), compared to 4.9 (*n* = 21) and 4.3 (*n* = 22) in 2018 and 2019, respectively. This difference reached statistical significance when comparing the years 2018 and 2019 with 2020 (*p* = 0.043 for 2018 vs. 2020, *p* = 0.012 for 2019 vs. 2020). During lockdown period I in 2020 11.1% of diagnosed melanomas were ulcerated, compared to 10.5% in post-lockdown period I and 13.5% in post-lockdown period II. The percentages of ulcerated melanomas in 2018/2019 and 2020 are depicted in Figure 4.

## 4. Discussion

Our data revealed a marked decline in melanoma diagnoses during certain periods of 2020—the first year of the COVID-19 pandemic. Compared to previous years, these differences were particularly pronounced during the first lockdown period. This observation is consistent with other reports describing a clear drop of patient encounters in all levels of the health care system [3]. Marson et al. reported that the number of newly diagnosed cutaneous melanomas dropped by 43.1% during the COVID-19 peak in 2020 (compared to 2019) [9]. The changes observed during the lockdown period by these authors are comparable to our results, with a 44% and 58% drop in melanoma diagnoses compared to 2018 and 2019, respectively.

Regarding a potential impact of the implemented lockdown measures on prognostic traits of primary melanomas such as tumor thickness and ulceration, we only observed a subtle increase in the mean Breslow thickness after the first COVID-19 lockdown in 2020. However, we detected a significant increase in the number and proportion of ulcerated melanomas in 2020 compared to previous years. The adverse prognosis associated with ulceration of primary melanomas has been linked to distinct immunological changes, which in turn potentially affect efficacy of adjuvant immunotherapy later in the course of the disease [10].

Consistent with our findings, an Italian study [6] showed a distinct increase in the mean Breslow thickness after the first COVID-19 lockdown in 2020 (0.66 mm during the lockdown period and 1.96 mm during the post-lockdown period, respectively) as well as in the percentage of ulcerated primary tumors (8.3% during the lockdown compared to 23.5% during the post-lockdown period) [6]. The less pronounced corresponding differences in our study sample may result from the fact that Italy was the European country most severely affected by the virus outbreak in early 2020, leading to more rigorous governmental restrictions and related avoidance of hospital or medical office visits. Nevertheless, the delay in melanoma diagnoses paired with an increased frequency of ulcerated primary tumors observed in our cohort still may adversely affect clinical outcomes in the long term [4].

In this context, we additionally noted that fewer sentinel lymph node biopsies for melanoma were performed at our department in 2020. This also reflects a pandemic-driven redistribution of hospital resources limiting capacities for surgical interventions.

For unknown reasons, we observed a significantly higher number of primary melanoma diagnoses in the first period of 2020 (pre-lockdown period I). As a result, the difference in total melanoma diagnoses between 2020 and previous years may have been attenuated.

In conclusion, our results show that governmentally enforced measures to control the COVID-19 pandemic clearly impacted on the diagnosis of primary melanoma. As both quantity and prognostic features of diagnosed melanomas were altered during and shortly after the first lockdown period, these changes most likely resulted from delayed office visits and cancelled skin cancer screening appointments. Continuous monitoring of the present and other “COVID-19 melanoma patient cohorts” will be necessary to better determine a potential adverse impact on long-term mortality from melanoma in these patients.

Although the novelty of our findings is limited, as similar developments have been described in other countries, we consider publication of national data to be important in this context, since the dimension of COVID-19-related lockdown measures varied between countries and national experience from the first lockdown in 2020 may guide public decision-making in the future. In general, for potential upcoming pandemic situations, health authorities should be advised that immediate death rates from the infectious disease per se have to be considered in the context of concomitant collateral damage caused by containment measures, such as increased long-term death rates from cancer. Thus, even in a pandemic, routine medical office visits, e.g., skin cancer screening appointments, should not be restricted, but permitted in compliance with specific anti-viral precautionary measures.

## Figures and Tables

**Figure 1 jcm-10-04059-f001:**
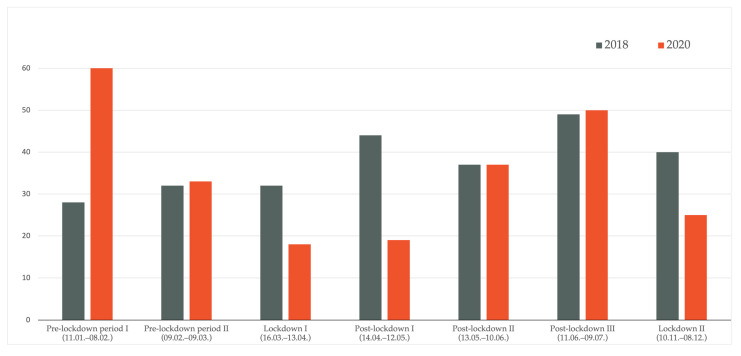
Comparison of melanoma cases 2018 versus 2020. A significant difference was detected for the first pre-lockdown period I (11 January–8 February), lockdown period I (16 March–13 April), and post-lockdown period I (14 April–12 May).

**Figure 2 jcm-10-04059-f002:**
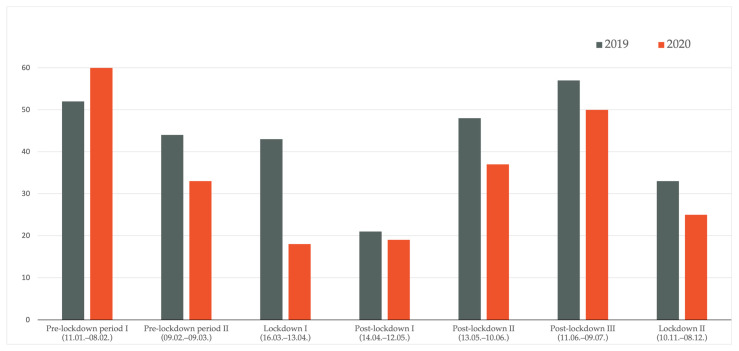
Comparison of melanoma cases 2019 versus 2020. Significant differences were detected in the lockdown period I (16 March–13 April).

**Figure 3 jcm-10-04059-f003:**
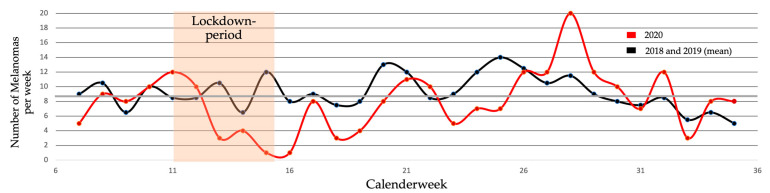
Number of diagnosed primary melanomas per week (mean of 2018 and 2019 compared to 2020). A marked decline in weekly melanoma diagnoses can be observed shortly after governmental lockdown measures were implemented in March 2020.

**Figure 4 jcm-10-04059-f004:**
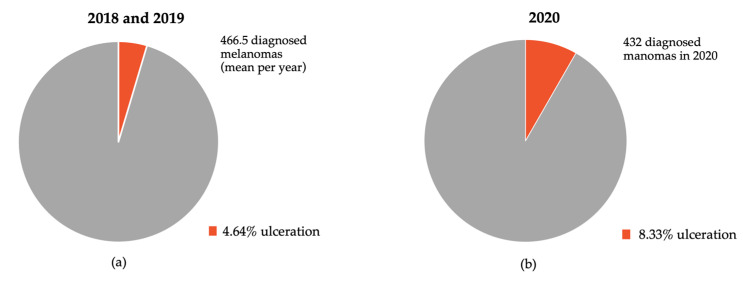
Percentage of ulcerated melanomas diagnosed in (**a**) 2018/2019 and (**b**) 2020 depicted in red; 4.9%, 4.3%, and 8.3% of all melanomas diagnosed in 2018, 2019 and 2020 were ulcerated (*p* = 0.043 for 2018 vs. 2020, *p* = 0.012 for 2019 vs. 2020).

**Table 1 jcm-10-04059-t001:** Primary melanoma diagnoses from 2018 to 2020: patient and tumor characteristics.

Time Period	Number of Diagnosed MM (*n*/% of Total Per Year)	Median Breslow (mm)	Number of Ulcerated MM (*n*/% of Total MM in This Period)	Number Female Patients (*n*/% of Total MM in This Period)	Number Male Patients (*n*/% of Diagnosed MM)	Median Age (Years)
2018	428	0.65	21 (4.91%)	200 (46.73%)	228 (53.27%)	61
16 March–13 April 2018 (LD)	32 (7.48)	0.69	0 (0)	14 (43.75)	18 (56.25)	60
14 April–12 May 2018 (P-LD1)	44 (10.28)	0.62	4 (9.09)	20 (45.45)	24 (54.55)	60
13 May–10 June 2018 (P-LD2)	37 (8.64)	0.6	0 (0)	20 (54.05)	17 (45.95)	57
2019	505	0.60	22 (4.36%)	245 (48.51%)	260 (51.49%)	60
16 March–13 April 2019(LD)	43 (8.51)	0.58	3 (6.98)	25 (58.14)	18 (41.86)	63
14 April–12 May 2019(P-LD1)	21(4.16)	0.65	1 (4.76)	14 (66.67)	7 (33.33)	60
13 May–10 June 2019(P-LD2)	48 (9.5)	0.77	2 (4.17)	24 (50)	24 (50)	63
2020	432	0.70	36 (8.33%)	199 (46.06%)	233 (53.94%)	63
16 March–13 April 2020(LD)	18 (4.17)	0.76	2 (18)	8 (44.44)	10 (55.56)	62
14 April–12 May 2020(P-LD1)	19 (4.4)	0.83	2 (10.53)	7 (36.84)	12 (63.16)	61
13 May–10 June 2020(P-LD2)	37 (10.88)	0.65	5 (13.51)	18 (48.65)	19 (51.35)	66

LD = lockdown, MM = malignant melanoma, P-LD1 = post-lockdown period I, P-LD2 = post-lockdown period II.

## Data Availability

The data presented in this study are available on request from the corresponding author.

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
