# Peer review of "COVID-19-Induced Reduction in Primary Melanoma Diagnoses: Experience from a Dermatopathology Referral Center"

_jcm, 2021, doi:10.3390/jcm10184059_

Round 1
Reviewer 1 Report
Dear authors, your article is a retrospective analysis of melanoma diagnosis in a referral center during the Corona-pandemic. It is well written and structured. There have already been other publications to this topic so I miss a little the novelty. Evaluation of data from several centers in Austria would have provided an even more valid insight into incidence during and after lockdown.
Reviewer 2 Report
The authors have conducted an interesting study examining if national measures taken during the Covid-19 pandemic were having an effect on the diagnostic features of primary melanoma in an austrian cohort. For that reason they conducted a restrospectively analysis of numbers and histologic features of primary melanoma (i.e. Breslow thickness, ulceration) from different time periods in 2020 (two pre-lockdown-periods: 11.01.-08.02. and 09.02.-08.03., two lockdown-periods: 09.03.-13.04. and 11.11.-08.12., two post-lockdown-periods: 14.04.-12.05. and 13.05.-10.06.) and compared them with previous corresponding time spans in 2018 and 2019. They showed a significant decline in the number of diagnsed melanomas especially for the first lockdown period in March/April 2020 compared to the two preceding years, followed by a similar low amount in the post-lockdown-period I. Interestingly, these time periods were followed by an distinct, but not-explicable rise in the post-lockdown-period II. Altough, the mean Breslow thickness in the first lockdown-period showed a slight upward trend, the numbers were not statistically significant. However, this study highlighted similar results regarding the ulceration status of detected primary melanoma during the two post-lockdown-periods (2020 compared with 2018 and 2019), resembling other already published studies in this field.
To sum up, this study adds more important national data, which can be crucially helpful to determine future containment measures in subsequent pandemic scenarios.
